# Sex-Specific Electrocortical Interactions in a Color Recognition Task in Men and Women with Opioid Use Disorder

**DOI:** 10.3390/biomedicines13123002

**Published:** 2025-12-08

**Authors:** Jo Ann Petrie, Abhishek Trikha, Hope L. Lundberg, Kyle B. Bills, Preston K. Manwaring, J. Daniel Obray, Daniel N. Adams, Bruce L. Brown, Donovan E. Fleming, Scott C. Steffensen

**Affiliations:** 1Department of Psychology, Brigham Young University, Provo, UT 84602, USA; petriejoann@gmail.com (J.A.P.); abhitris@gmail.com (A.T.); hlundber@gmail.com (H.L.L.); kbbills@noordacom.org (K.B.B.); obray@musc.edu (J.D.O.); bruceleonardbrown@gmail.com (B.L.B.); flemingdr16@gmail.com (D.E.F.); 2Department of Neuroscience, Noorda College of Osteopathic Medicine, Provo, UT 84606, USA; 3Department of Electrical Engineering, Brigham Young University, Provo, UT 84602, USA; preston_manwaring@byu.edu; 4Photopharmics, Inc., 3450 Triumph Blvd, Lehi, UT 84043, USA; dan.adams@photopharmics.com; 5Department of Neurosurgery, Brain and Spine, University of South Florida, Tampa, FL 33612, USA

**Keywords:** dopamine (DA), sex-related differences, electroencephalogram (EEG), substance use disorder (SUD), opioid use disorder (OUD), event-related potentials (ERP), visual attention, visual evoked potential (VEP), color vision processing

## Abstract

**Background**: Opioid use disorder (OUD) and associated overdose deaths have reached epidemic proportions worldwide over the past two decades, with death rates for men consistently reported at twice the rate for women. We have recently reported sex-specific differences in electrocortical activity in persons with OUD in a visual object recognition task. The mesolimbic dopamine (DA) system is implicated in OUD but also plays a critical role in some disorders of visual attention and a modulatory role in the processing of visual stimuli in the blue cone pathway of the retina. We hypothesized that electrocortical responses to color stimuli would be affected differentially in men and women with OUD. **Methods**: Using a controlled, cross-sectional, age-matched (18–56 years) design, we evaluated color processing in male and female subjects recruited from a community-based, high-intensity residential substance abuse and detoxification treatment program. We evaluated electroencephalogram (EEG) event-related potentials (ERPs) and reaction time (RT), in male and female participants with OUD (*n* = 38) vs. sex- and age-matched non-OUD control participants (*n* = 37) in a simple color recognition Go/No-Go task, as well as perceptual and behavioral responses in physiological and neuropsychological tests. **Results**: N200, P300, and late potential (LP) *Relevant* stimulus-induced ERPs were evoked by the task and were well-differentiated from *Irrelevant* distractor stimuli. P300 amplitudes were significantly greater and N200 and LP latencies were significantly shorter in male vs. female non-OUD controls in this task. There were significant sex differences in N200, P300, and LP amplitudes and latencies between male vs. female non-OUD subjects and OUD differences with blue color as the Relevant stimulus. In the *Binocular Rivalry Test*, there were shorter dwell times for perceiving a blue stimulus in male OUD subjects and there were significant sex and OUD differences in neuropsychological tests including *Finger Tapping*, *Trails A/B*, and *Symbol Digit Modalities Test*. **Conclusions**: These findings suggest that there are significant sex-related physiological, perceptual, and cognitive differences in color processing that may result from deficits in DA production in the retina that mirror deficits in mesolimbic DA transmission correlating with OUD, suggesting that blue color processing has the potential to be an effective biomarker for brain DA and for diagnosis and monitoring of treatment efficacy in substance use disorders.

## 1. Introduction

### 1.1. Background and Significance

The personal and socioeconomic consequences of substance use disorders (SUDs) and addiction are staggering and have proven devastating to human society worldwide. The National Institute on Drug Abuse’s (NIDA) 2024 Annual Fact Sheet [1] using postmortem data from the U.S. Centers for Disease Control and Prevention’s (CDC) public health database collection termed CDC WONDER (for more information see https://wonder.cdc.gov/wonder/help/main.html#WhatisWONDER (accessed on 7 January 2025) reported that in 2020 approximately 40.3 million people, 12 years and older, in the USA reported having an SUD, but only 6.5% received treatment, and in 2021 more than 107,000 people in the USA alone died from drug-involved fatal overdoses, including illicit drugs and prescription opioids—mostly from fentanyl poisoning [2]. In addition, in 2023 Butelman et al. [3] reviewed these overdose deaths at the state-level from 1999 to 2023 and found that younger men (25–64 years old) have consistently been proven to be more vulnerable and at risk for overdose to opioids and stimulant drugs at two to three times the rate of women of the same ages. This remained true even when adjusting for environmental and socioeconomic variables and has since called for an immediate united focus in research for the “biological, behavioral, and social factors that underlie sex differences in human vulnerability to drug overdose”. It is of particular concern in the USA among women and men with opioid use disorder (OUD) and overdose deaths, where men have two to three times the mortality rate from OUD than women.

### 1.2. Biomarkers for Dopamine in Substance Abuse Disorders

The dogma in the addiction field is that dopamine (DA) neurotransmission in the pleasure/reward center of the brain is enhanced by acute drug and alcohol consumption but then is diminished by chronic use [4,5,6,7]. In 2003, Elkashef and Vocci [8] posited that the discovery of a reliable biological marker or trait for addiction would be a major breakthrough. State markers would be extremely helpful if they change consistently in relation to the drug dependency state distinguishing acute vs. chronic effects, or in the abstinence state distinguishing between acute withdrawal and the more prolonged “craving” state correlating with further relapse. An important component for any therapeutic individual multimodal addiction treatment plan would then be to identify specific biological changes that could be used to monitor and tailor treatment objectively [9,10,11]. In addition, a reliable biological marker of addiction might be helpful for the following: (1) targeting specific pharmacological agents for subgroups of addicts; (2) determining the length of treatment evidenced to be beneficial; (3) preventing relapse by initiating treatment at the earliest sign of an increase or re-emergence of a certain marker; and (4) understanding traits that might predict or indicate high risk for opioid use, misuse, dependence, and addiction. Monitoring DA transmission is a valid way to objectively determine potential sex-specific differences for intoxication and addiction. For over three decades, DA, its receptors, transporters, precursors, and metabolites in the mesolimbic reward system have been explored as potential biomarkers of substance use and addiction [4,12,13,14,15,16,17,18,19,20]. However, there are no reliable and non-invasive biomarkers of addiction currently in clinical use. This lack of an objective index of the addictive state has stalled innovations in treatment strategies.

### 1.3. Electrocortical Activity in Opioid Use Disorder

Electrophysiological (EEG) scalp recordings, such as event-related potentials (ERP) and visual event potentials (VEPs), are recognized as a non-invasive objective neuroimaging techniques for studying neuronal information processing associated with normal and abnormal cognition [21,22], including those found in long-term substance use. For example, P300 ERP assessment of detoxification effects on cognition in opioid and cocaine users are ameliorated by the partial mu-opioid receptor partial agonist buprenorphine [23]. Particular to the current study, the P300 event-related potential (ERP) has shown to be an index of human cognition, providing an objective measure of brain activity that is very sensitive to any central nervous system (CNS) disruptions and pharmacologic manipulations [24]—including possible biological and/or genetic underlying mechanisms of brain oscillations that drive the ERPs [25]. For example, decreased P300 amplitudes in acute ethanol and marijuana use, with no consistent differences in latencies, has been recognized for over 35 years [26,27]; while a similar significant reduction in P300 amplitudes in *chronic* alcohol abuse is known to be coupled with increased latencies [28,29]. These findings are similar to the few studies in opioid addiction, with a similar showing of decreased P300 amplitudes and increased latencies [23,30,31,32].

### 1.4. Rationale and Hypotheses

We have recently reported gender-specific differences in ERPs and wavelets in a simple visual object recognition task in those individuals with OUD [33] in an effort to better elucidate why men have such a higher mortality rate in SUDs and addiction. The current study extends the results of this study and others [34,35,36] to determine if there might also be significant sex differences in ERPs in a simple color recognition Go/No-Go task. Based on our previous ERP study, wherein we found that non-OUD males had greater N200 and P300 amplitudes and shorter N200 and P300 latencies than non-OUD females [33], we hypothesized that males would exhibit similar ERP effects in a simple color recognition task. As men exhibit higher rates of opioid abuse, but women may develop dependence more quickly and are more susceptible to the addictive nature of opioids [37,38], we hypothesized that there would be significant sex differences in color processing, in particular blue color processing in the retina which is influenced by DA levels via indirect pathway DAergic amacrine cell modulation of the direct blue cone color pathway [39]. Understanding the unique differences men and women face in addiction, treatment, and recovery that may be due to distinctive differences in physiology, hormones, mental health, and/or life circumstance, any biomarkers for brain DA would allow for objective monitoring of the addictive state, the efficacy of treatment, and the potential to enhance recovery and prevent relapse.

## 2. Methods

### 2.1. Ethics Statement

Prior approvals and permissions were received from the Director of the addiction treatment program where all substance abuse participants were recruited. It should be noted that as a publicly funded facility in Provo, UT, USA, the participating treatment center was under the Confidentiality of Substance Use Disorder Patient Records Law, (i.e., Code of Federal Regulations (C.F.R.), Title 42, Part 2, 2010) and required to meet very stringent confidentiality and HIPAA regulations (i.e., Health Insurance Portability and Accountability Act of 1996, Pub. L. No. 104-191, S. 264). All data were handled with confidentiality and with compliance to HIPAA regulations. Signed consent forms were received before any testing started and any amendments or changes in protocols were met with approval from the Brigham Young University (BYU) Institutional Review Board (IRB), Provo, UT, USA, (F100210; Approved 10 January 2012). Strict adherence to ethical issues and research standards were maintained.

### 2.2. Participants

There were ninety-eight initial participants in this study made up of four groups: Male non-OUD controls (*n* = 30, age range = 17–53 years); female non-OUD controls (*n* = 30, age range = 18–56 years); male OUD (*n* = 19, age range = 18–53 years); and female OUD (*n* = 19, age range = 18–51 years). It should be noted that for the purposes of this study the use of the term’s “women” and/or “men” or “female” and/or “male” are interchangeable and refer to the biological sex and/or gender that our participants identified as at the time of this study. Nineteen female and nineteen male OUD participants were recruited from a community-based high intensity residential substance abuse and detoxification treatment program. This is a state-funded, residential, short-term (30 days or less), and long-term (more than 30 days) detoxification and addiction treatment center for treatment of persons with or without co-occurring mental and substance abuse disorders which provides a vertically integrated continuum of care according to the most current published American Society of Addiction Medicine (ASAM) Patient Placement Criteria (for further information see https://www.asam.org/asam-criteria (accessed on 25 July 2025)). The therapists, counselors and intake staff were contacted and were provided with an electronic copy of the IRB protocol, a participant recruitment flyer for inclusion or exclusion, and a pre-screen checklist to be reviewed by the therapist with those clients interested in participating in the research project. Inclusion criteria were as follows: men and women in the OUD group were age 18–55 years, still experiencing daily drug cravings, no prior history of severe brain injury, seizure free, not prone to fainting, not claustrophobic, willing to indicate all drug use on drug history sheet, to not have used opiates in the past 24 h but were no longer than 60 days out, willing to give a sample for urinalysis (UA), willing to spend three hours being escorted to and from the BYU EEG facility with two hours of neuropsychological and EEG testing, and willing to keep confidential all proceedings and take offered compensation—$20.00 Visa gift card and a packaged food item.

The non-OUD control groups were matched as closely for age and gender as possible with the OUD group. After individually looking at the neurophysiological testing data collected from all participants related to the current study, the data from seventy-five were retained for inclusion in the final analyses. Approximately twenty-seven percent of the EEG data collected were found to have recording discrepancies giving a final sample size for analyses of 75 participants. Participants were screened to ensure they were in good overall health, with no personal history of physiological or psychological disorders, and/or head trauma, and were not taking any medication. Inclusion criteria for the non-OUD control group participants were as follows: drug-free men and\or women, age 18–55 years, no prior history of severe brain injury, seizure free, not prone to fainting, not claustrophobic, willing to indicate all drug use on drug history sheet, to not have used any opioid pain killers in the last three months or sleep aids in the past month or any other mood altering prescription drugs, willing to give a urine sample for UA, willing to spend 2–3 h participating in neuropsychological and EEG testing, willing to keep confidential all proceedings and take offered compensation (e.g., extra credit for a psychology or neuroscience class) and a packaged food item. After a complete description of the study to the participants, written informed consent was obtained from all participants.

Each potential participant was asked to fill out an on-site computerized questionnaire prior to the EEG session to determine whether they met study qualifications. Once pre-qualification criteria were met and demographics information recorded, participants were immediately scheduled for an EEG session, informed of the complete description of the study by well-trained assistants, asked if they were willing to participate, and then written consent was obtained. In order to verify that every participant was truly drug-free, all participants that were specifically age- and gender-matched to the OUD group were given the same UA test that is normally given randomly in the treatment program to check for compliance to their programs [40]—all others completed the drug disclosure questionnaire. The UA tested for a variety of illicit drugs including opiates, marijuana, cocaine, benzodiazepines, and psychostimulants. All OUD and non-OUD controls participants were tested, with all UA samples collected prior to testing. The same type of testing kit was used for all. The UAs were de-identified, frozen immediately, and taken within 48 h to be tested at the treatment program by a stringent Research Institute on Addictions (RIA; University of Buffalo, New York) analysis. Each UA sample was labeled with a coded identifying number plus with the date of the sample to identify it. The participants were advised of their confidentiality rights and took all of the other tests at the same time the UA sample was given. A positive UA was to result in disqualifying the subject data due to drugs in the system—no assessment was discarded because of a positive UA. If the participant was unable to give the required sample at the beginning, we would ask regarding it at various times of the testing so a good undiluted sample could be given. In addition, at this time all participants (non-OUD controls vs. OUD) completed a drug screening questionnaire listing all drugs used, amounts, when last used, and when drug abuse started as applicable.

### 2.3. Electroencephalographic Evoked-Potential Recordings with a Dense Array Sensor Net

A 128-channel sensor net design was used to acquire dense array EEG data; it had sponge inserts in each electrode pedestal allowing the electrolyte saline solution to be held for 2 h recordings. The Geodesic EEG System (GES) HydroCel Geodesic Sensor Net (HCGSN; Electrical Geodesics, Inc., Eugene, OR, USA) is specifically designed to give surface tension and even distribution of the electrodes across the spherical surface of the head. The networking on an HCGSN adjusts the spatial location of each sensor until a single distance spans all pairs along direct lines, each line being a *geodesic* (i.e., the shortest distance between two points on the surface of a sphere) for the active and grounded sites. An accurate geodesic tessellation of the head surface optimizes the sampling of the electrical field. The internationally accepted 10–20 placement system of the sensor net’s electrodes (electrodes are placed at 10% and 20% along lines of latitude and longitude) was used for montage points of the Average-Reference Mastoid Montage established by the *International Federation of Clinical Neurophysiology* [41]. The participant’s head was measured to determine the size of the sensor net needed. This was accomplished by measuring the circumference of the head from ear to ear around the head. The head was measured from ear tragus to ear tragus (small protruding bumps in front of ears) over the top of the head with a grease pen used to mark the half-way spot on the top of the head. The head was then measured from nasion to the inion and the scalp was again marked where it crossed the other ear to ear mark on top of the head. This established the center of the head where the Cz REF electrode on the sensor net (center of net) was to be placed to obtain good spherical tension and proper signal reading. The sensor net was soaking in a special potassium chloride solution and baby shampoo and water solution for improved electrical conduction while this was being performed—2–3 min. For comfort purposes, the participant was also given a towel and a washcloth to help keep from getting wet and cold from the saline solution and asked to close their eyes while the net was placed (two people placed the net allowing for better fit).

VEP’s were acquired in 1 s epochs of stimuli for each visual stimulus presentation. E-prime 3.0 software (Psychology Software Tools, Inc., Sharpsburg, PA, USA) was used for the visual attention tasks developed. The net was then plugged into the amplifier and the EGI and E-prime software connections were established and any impedance-to-current-flow was checked until it was reduced to 10 Ohms (10 kΩ) at each electrode. Impedance measures were checked between each session to account for any evaporation of electrolyte solution during the computer testing. Participants were asked to come with freshly washed hair with no conditioners or gels in order to ensure good conductivity. Participants were shown how wiggling in the seat, nose scratching, clenching their teeth, blinking their eyes a lot, etc., could change the output of the data on the EEG and how sensitive the sensor net was to any component generation of unnecessary neural “noise” from body movement. When each task began, the participant read a standard set of instructions explaining the task and a sample visual stimulus was shown on the computer screen. Reaction times (RT) were measured from the time the stimulus was presented until the participant pressed the button. An incorrect response notice appeared on the screen if participant pressed the button when no target color was on the screen.

### 2.4. Color Processing Task

The color processing task used to induce ERPs was a “Go/No-Go cognitive test.” This task consisted of three separate sessions or blocks of four min duration each of flashing Red, Green, and/or Blue colors that filled the full computer monitor. Participants were asked to respond via key press when they detected the Relevant stimuli (either Blue, Red, or Green as instructed in the programing of each session—no one knew what color was to be Relevant at any given time including research assistants or senior investigators). The stimuli were randomly presented (50 ms duration) on a computer monitor and participants were instructed to not respond when Irrelevant color flashes were shown. For example, if instructions were given to respond to Blue (Relevant stimulus) when presented, then the participant was to key press as quickly as possible (Go) but to ignore and not respond (No-Go) to the Red and/or Green (Irrelevant stimuli) flashes. E-Prime 3.0 software (Psychology Software Tools, Inc., Sharpsburg, PA, USA) was used to run the visual attention task and the stimuli were presented on a PC-type computer screen/monitor with high density (HD) 1080p LED resolution. The Geodesic EEG System 300 was used to record EEG activity and participants wore a 128-channel sensor net during each session—sized to fit their individual head size. The Geodesic system is programmed to monitor any EEG artifacts automatically and flagged potential artifacts associated with excessive head, muscle, or eye-blink movements. Evoked potentials were acquired in 2 s epochs during each visual stimulus presentation; they began 100 ms prior to and ended 900 ms after each stimulus presentation. The EEG data around each visual stimulus were averaged to obtain the visual evoked potentials (VEPs) from all participants.

The averaged VEP consisted of multiple components which were identified by their respective positions on the waveform, relative to the time of stimulus presentation. Eight distinct alternating positive/negative peaks on the VEP waveform were identified, which occurred at characteristic latencies from the time of stimulus presentation. Early and late peaks of the VEP were identified according to established convention and were labeled N50, P100, N100, P200, N200, and P300, respectively. While the early components (i.e., N50, P100, N100, and P200) of the averaged VEP waveforms were relatively unaffected by the type of visual stimulus presented, the late components of the averaged VEP waveforms (N200, P300, and Late Positive) evidenced significant amplitude differences across conditions in this simple Go/No-Go task. These late components are termed event-related potentials (ERPs). Instructions were exactly the same for each color when Relevant. This was designed to compare the ERPs for each color wavelength responded to when either Relevant or Irrelevant. Each session began with a block of three practice presentations followed by 40 test presentations of each of the three stimuli, randomized by condition, and interval.

### 2.5. General Behavioral Procedures

All participants were given the same battery of pre-screening checks, questionnaires, neuropsychological tests, computer testing, color tests, and measures of handedness and eye dominance. Cognitive behavioral testing was performed by standardized “pen and paper” neuropsychological tests. Reaction time and physiological responses were measured with EEG and recorded during a battery of computer screen Go/No-Go tasks. The battery of three cognitive, physiological, and behavioral tasks (i.e., the *Finger Tapping Test*, the *Symbol Digits Modality Test*, and the *Trails A/B* tests) was used to provide the necessary scientific rigor to evaluate the utility of color processing as an objective index of the addictive state. All participants were asked to participate in a single session of neuropsychological testing and a separate physiological recording session; all were presented with the same tasks. Each participant was then checked for handedness, eye dominance, and age, and then tested for any visual color deficiency.

### 2.6. Hardy-Rand-Rittler (HRR) Pseudoisochromatic Test

Each participant was administered the HRR visual color test—this was a 5 min color deficiency test given prior to having the EEG net placed at the recording session to determine the extent of any color deficiency. The test is a series of color plates shown to the participant who responds to certain questions of “How many?,” “What?,” and “Where?” of any symbols seen on each plate (given in varying degrees of color embedded in a grayscale background) by verbal response and by tracing with a brush any symbols seen—no one with a color deficiency was included in the testing.

### 2.7. Finger Tapping Test

This is a straight-forward neuropsychological test used to assess motor speed and motor control (especially in those who have experienced some type of brain injury) and is a simple finger tapping test that can measure and compare the participants’ reaction times (RTs). Participants placed their dominant hand palm down, fingers extended, with the index finger resting on a lever that was attached to a counting device. Individuals were instructed to tap their index finger as quickly as possible for 10 s per trial, keeping the hand and arm stationary and were not allowed to hold the hand down. This trial was repeated five times on each hand, with hand changing every two times per hand until five trials per hand were completed—this change was performed to avoid great hand fatigue. Before starting the test, individuals were given a practice session.

### 2.8. Symbol Digit Modalities Test© (SDMT)

The SDMT© involves a simple substitution task that typically developing children and adults can easily perform. Using a reference key, the examinee was given 90 s to pair specific numbers with given geometric figures. Responses were written but could be orally given, if necessary. For either response mode, administration time took 5 min with simplified scoring. Individuals with any cerebral dysfunction perform poorly on the SDMT©, in spite of normal or above average intelligence. This test was used because it has been proven for its effectiveness in a wide range of clinical applications, including the following: head injuries; strokes; brain tumors; reading difficulties; learning disorders; Alzheimer’s disease; viral, bacterial, and other cerebral infections; pre-, peri-, and early postnatal insults; senile dementia; aphasia; neurotoxicity; alcoholism; cerebral anoxia; Huntington’s Disease; and so forth.

### 2.9. Trail Making Test, Parts A or B (Trails A/B)

The *Trail Making Test* is a neuropsychological test of visual attention and task switching which was randomized by giving one-half of the participants Part A, while the other half were given Part B to allow for a second testing if necessary and no test-retest confounds. This cognitive test consists of two parts and provides information about visual search speed, scanning, speed of processing, mental flexibility, as well as executive functioning. If the participant made mistakes, the mistakes were quickly brought to their attention, time was stopped, and then they continued from the last correct circle. This test took approximately 5 to 10 min to complete. This test was originally known as Partingon’s Pathways, or the Divided Attention Test, which was part of the Army Individual Test Battery and evaluates information processing speed, visual scanning ability, integration of visual and motor functions, letter and number recognition and sequencing, and the ability to maintain two different trains of thought. The test can be administered orally if an individual is incapable of writing. Poor performance is associated with many types of brain impairment, in particular frontal lobe lesions.

### 2.10. Binocular Rivalry Test

Participants donned a pair of 3D anaglyph glasses with a red gel lens on one side and a blue gel lens on the other positioned so that the red lens was over the dominant eye in order to control for dominance. They were advised to gaze at a static picture on the computer screen consisting of superimposed images of the number “1” in red and the number “2” in blue. The red lens presented the red “1” to one eye and the blue “2” to the other. Due to binocular rivalry, only one of the images was perceived at a time and the perception alternated approximately 2–5 s in non-OUD subjects. The participant was advised to fixate on the computer screen with their fingers on the keys 1 and 2 on the number pad on the computer keyboard. They were instructed to press the number “1” when it was perceived and the number “2” when it was perceived. Key presses were captured.

### 2.11. Statistical Analyses

The EEG waveforms evoked by the stimuli were averaged to obtain the VEP for each participant. At each of the 128 electrodes, the visual presentations were averaged within each subject and between conditions (e.g., Relevant vs. Irrelevant stimuli) and then grand-averaged within groups. Amplitude and latency were measured for each peak of the within-subject average VEP components for the N100, P100, N200, P200, P300, late negative, or N400, and late positive (LP) or P700 waveforms, using NetStation^®^ Data Analysis Tools (Electrical Geodesics, Inc., [EGI], Eugene, OR, USA) with an adaptive means algorithm. To simplify the analysis, we focused on the 18 electrodes of the 10–20 International electrode system. To further simplify, we consolidated the 18 electrodes into groups of anterior, central, and posterior or front, middle, and back electrode locations on the head. We chose to evaluate averaged posterior electrodes of the 10–20 international electrode system (T5, P3, Pz, P4, T6, O1, and O2) for measurements of N200 and P300 and averaged anterior electrodes (FP1, FP2, F7, F3, Fz, F4, and F8) for measurements of LP, as in our previous report on ERPs in subjects with OUD in an object recognition task [33]. The quantitative electrophysiological data obtained through the EGI Data Analysis Tools were analyzed using the SAS/STAT^®^ Proc Mixed 9.2 statistical analysis program [42]. This statistical program was meant to fit multilevel and hierarchical linear models and considered suitable for the “mixed” statistical data from the current study; data on individuals were nested within naturally occurring hierarchies such as men and women within addiction. It was also considered that a multivariate analysis was not necessary since there was a great deal of significance found in the analyses associated with all hypotheses. Igor^®^ Pro v9 software (WaveMetrics, Lake Oswego, OR, USA) and Excel (Microsoft Office) were used to present the data in graph and layout form for easier reading. Measures of Reaction Times (RT) were analyzed with ANOVA.

## 3. Results

### 3.1. Sex Differences in Event-Related Potentials (ERPs) in a Color Recognition Go/No-Go Task in Non-OUD Control Subjects

We compared VEPs elicited by Red, Green, and Blue stimuli superimposed in men and women non-OUD subjects in the color recognition task at electrodes Pz (N200 and P300, Figure 1A) and Fz (LP, Figure 1B). N200 and P300s were best differentiated at the back of the head and LPs at the front of the head in association with the Relevant stimulus, as shown in the topomaps for all 128 sensors on the head. The amplitude of the P300 component of the waveform was much greater in association with the Relevant stimulus (Blue) than with Red and Green Irrelevant stimuli, in particular at occipital and parietal locations. While the grand-averaged ERP waveforms and topomaps demonstrated differences between responses obtained with the Relevant stimulus, averaging often underestimates the significance of the effects due to inter-subject temporal dispersion and other vagaries. Thus, we measured each ERP with individual measurements in each subject. These measurements were then submitted to statistical analysis. Figure 1E,F compares the distribution of ERP peak-to-peak amplitudes and latencies obtained at electrodes Pz (N200, P300; average of 7 electrode signals at the back of the head; see Methods) and Fz (LP; average of 7 electrode signals at the front of the head) in male and female non-OUD subjects (*n* = 22, 24 respectively). At the Pz electrode, males had significantly greater P300 amplitude than females (*F*_(1,45)_ = 12.2, *p* = 0.001; Male amplitude = 5.96 ± 0.70 mV vs. female amplitude = 3.12 ± 0.36 mV), but there were no sex differences for N200 amplitude (*F*_(1,45)_ = 0.115, *p* = 0.74; Male amplitude = 3.43 ± 0.41 mV vs. female amplitude = 3.22 ± 0.51 mV; Figure 1E). However, males had significantly shorter N200 latency than females (*F*_(1,45)_ = 19.3, *p* = 7.1 × 10^−5^; Male latency = 263 ± 4.22 ms vs. female latency = 295.5 ± 6.2 ms), but there were no sex differences for P300 latency (*F*_(1,45)_ = 1.2, *p* = 0.28; Male latency = 342 ± 8.1 ms vs. female latency = 328.4 ± 9.6 ms; Figure 1F). At the Fz electrode, there were no significant differences between males and females for LP amplitude (*F*_(1,45)_ = 3.56, *p* = 0.07; Male amplitude = 7.33 ± 0.76 mV vs. female amplitude = 5.32 ± 0.75 mV; Figure 1E). Males had significantly shorter LP latencies than females (*F*_(1,45)_ = 4.73, *p* = 0.03; Male latency = 679 ± 13.8 ms vs. female latency = 712.9 ± 11.1 ms; Figure 1F).

### 3.2. Color Differences in Event-Related Potentials in a Color Recognition Task in Non-OUD Control Subjects

We hypothesized that blue color processing in this color recognition cognitive task would be modulated by sex in non-OUD controls. We wanted to determine if color wavelength was a factor for the Relevant stimulus. Thus, in addition to Blue as the Relevant stimulus (Figure 1), we compared the effects of Red and Green color presentations as Relevant stimuli in the same Go/No-Go task in non-OUD controls. Figure 2 shows superimposed grand-averaged waveforms and topomaps obtained at Pz (P300; Figure 2A,B) and Fz (LP; Figure 2C,D) for Red, Green, and Blue as Relevant stimuli in separate sessions in male and female non-OUD controls. There were no obvious differences in N200, P300, or LP amplitudes based on grand-averaged waveforms. However, as mentioned above, grand-averaged waveforms are only suggestive and tend to underestimate effects. Thus, ERPs were measured in each subject and then submitted to analysis. We ran a SAS/STAT^®^ Proc Mixed statistical analysis of ERPs obtained at electrodes in the front and the back of the head. Models were estimated using restricted maximum likelihood with Kenward-Rogers degrees of freedom with adjustments applied. For the N200, there were no effects of sex or color (main effect of female: *t*_(49)_ = −0.75, *p* = 0.455; main effect of Green: *t*_(50)_ = −0.63, *p* = 0.534; main effect of Blue: *t*_(50)_ = 0.92, *p* = 0.361; female × green interaction: *t*_(50)_ = 0.60, *p* = 0.554; female × Blue interaction: *t*_(51)_ = −0.63, *p* = 0.533; Figure 2E). For the P300, there was a significant interaction indicating reduced amplitudes for female participants on trials with a Blue target (female × Blue interaction: *t*_(50)_ = −2.79, *p* = 0.007, *b* = −2.2, 95% CI [−3.8, −0.6] µV; Figure 2F). No further significant effects of sex or color were uncovered (main effect of female: *t*_(49)_ = 0.12, *p* = 0.904; main effect of Green: *t*_(50)_ = 0.38, *p* = 0.706; main effect of Blue: *t*_(50)_ = 0.85, *p* = 0.401; female × Green interaction: *t*_(50)_ = 0.16, *p* = 0.875). For the LP, female participants displayed reduced amplitudes compared to males (main effect of female: *t*_(50)_ = −2.22, *p* = 0.031, *b* = −0.9, 95% CI [−1.8, −0.1] µV; Figure 2G), and amplitudes were significantly enhanced on trials with a Blue target across both sexes (main effect of blue: *t*_(50)_ = 5.85, *p* < 0.001, *b* = 2.5, 95% CI [1.7, 3.4] µV). There were no additional effects of sex or color (main effect of Green: *t*_(50)_ = −0.07, *p* = 0.945; female × Green interaction: *t*_(50)_ = 0.76, *p* = 0.451; female × Blue interaction: *t*_(51)_ = 0.21, *p* = 0.833). Table 1 shows ERP amplitudes for males vs. females for different Red, Green, and Blue as the Relevant stimulus and Table 2 shows significance levels for the various interactions between sexes.

### 3.3. Event-Related Potentials in a Color Recognition Task in OUD

Based on grand-averaged ERP waveforms and individual measurements (Figure 1), there appeared to be important sex differences in the color recognition task. Mainly, non-OUD male controls were typically characterized by greater N200, P300, and LP ERP amplitudes and shorter latencies. Moreover, based on wavelength studies (Figure 2), there were sex-related differences in Blue color cortical processing. One of the main objectives of this study was to evaluate cognitive processing in persons with OUD. In our previous ERP study using a simple object recognition task, N200, and P300 amplitudes were not significantly affected by OUD in males, but P300 amplitudes were decreased in females. However, latencies were affected differentially in male vs. female OUD subjects with males exhibiting longer N200 and P300 latencies and females exhibiting shorter N200 latencies compared to non-OUD controls [33]. Thus, we hypothesized that there would be similar deficits in cognitive processing in this simple color recognition task, but in particular for blue color processing given P300 and LP sex differences, in subjects with OUD. Thus, we compared ERPs in non-OUD controls vs. OUD participants in this color recognition task collapsed by sex.

Figure 3 compares grand-averaged P300 (Figure 3A) and LP (Figure 3B) ERP waveforms in male and female non-OUD vs. OUD subjects associated with Blue as the Relevant stimulus. There were obvious differences in ERP waveforms across groups. We measured each ERP with individual measurements in each subject. These measurements were then submitted to statistical analysis. Figure 3 compares the distribution of ERP peak-to-peak amplitudes (Figure 3C–E) and latencies (Figure 3F–H) obtained at electrodes Pz (N200, P300) and Fz (LP) in male and female non-OUD vs. OUD subjects. Regarding ERP amplitudes, there was no difference in male non-OUD vs. OUD N200 (*F*_(1,39)_ = 0.006, *p* = 0.94; Figure 3C), P300 (*F*_(1,39)_ = 0.42, *p* = 0.52; Figure 3D), or LP (*F*_(1,39)_ = 1.2, *p* = 0.28; Figure 3E) or female non-OUD N200 (*F*_(1,39)_ = 2.4, *p* = 0.12; Figure 3C), P300 (*F*_(1,39)_ = 0.09, *p* = 0.77; Figure 3D), or LP (*F*_(1,39)_ = 0.07, *p* = 0.78; Figure 3E) amplitudes in the color recognition task. However, there were significant differences between male vs. female OUD N200 (*F*_(1,29)_ = 3.9, *p* = 0.05; Figure 3C) and P300 (*F*_(1,29)_ = 5.5, *p* = 0.03; Figure 3D) amplitudes, but not LP (*F*_(1,29)_ = 0.73, *p* = 0.4; Figure 3E) amplitudes in this task. There was a difference in male non-OUD vs. OUD N200 (*F*_(1,39)_ = 3.92, *p* = 0.05; Figure 3F) and LP (*F*_(1,39)_ = 16.3, *p* = 0.0003; Figure 3H), but not for P300 (*F*_(1,39)_ = 0.002, *p* = 0.97; Figure 3G) latencies. While there was no difference in female non-OUD vs. OUD N200 (*F*_(1,39)_ = 0.11, *p* = 0.74; Figure 3F) or P300 (*F*_(1,39)_ = 0.03, *p* = 0.86; Figure 3G), there was a significant difference for LP (*F*_(1,35)_ = 5.99, *p* = 0.02; Figure 3H) latencies in the color recognition task. Most notably, while male OUD subjects had longer latencies than non-OUD male subjects, female OUD subjects had shorter latencies that non-OUD subjects. There were significant differences between male vs. female OUD N200 (*F*_(1,29)_ = 5.15, *p* = 0.03; Figure 3F) and LP (*F*_(1,29)_ = 17.7, *p* = 0.0002; Figure 3H) amplitudes, but not P300 (*F*_(1,29)_ = 0.47, *p* = 0.49; Figure 3G) latencies in this task.

Given these sex and OUD differences in color processing at Pz and Fz, we ran a SAS/STAT^®^ Proc Mixed statistical analysis of ERPs obtained at electrodes in the front and the back of the head, which was intended to fit multilevel and hierarchical linear models. The “mixed” statistical data from the current study were nested within naturally occurring hierarchies such as males and females and OUD. Table 3 lists the outcomes from that analysis. Proc Mix analysis found marked significance between male vs. female non-OUD subjects for N200 amplitudes and latencies, P300 amplitudes, and LP amplitudes and latencies, non-OUD vs. OUD subjects for LP latency, and male and female non-OUD vs. OUD subjects for LP latency.

### 3.4. Effects of Sex and OUD on Reaction Time in the Color Recognition Task

Reaction time (RT) was determined from the time of stimulus presentation to the time participants responded with a key press to the Relevant stimulus. In the color recognition task, there was no significant difference between males vs. females (*F*_(1,31)_ = 2.66, *p* = 0.11; male RT= 323.7 ± 10.1 ms; female RT = 348.5 ± 11.3 ms), nor between male non-OUD controls vs. OUD subjects (*F*_(1,32)_ = 0.94, *p* = 0.34; male OUD RT = 337.5 ± 9.9 ms), nor between female non-OUD controls vs. OUD subjects (*F*_(1,33)_ = 0.1, *p* = 0.75, female OUD RT = 352.9 ± 7.9 ms).

### 3.5. Color Differences in Binocular Rivalry in OUD

Given the sex differences in color processing, we hypothesized that physiological tests involving Blue color vision would be sex-dependent and affected in subjects with OUD. We chose the *Binocular Rivalry Test* to evaluate sex and OUD differences between Blue vs. Red color perception. Perceptual alternation rates/min in the *Binocular Rivalry Test* were as follows: male controls: 19.4 ± 2.07 (*n* = 15); male OUD: 18.3 ± 2.1 (*n* = 15); female controls: 21.1 ± 2.5 (*n* = 17); and female OUD: 23.2 ± 3.2 (*n* = 19) per min. There was no significant difference in alternation rates by sex or OUD (*p* > 0.05). Thus, we evaluated the dwell time for each of the stimuli (Figure 4), which we hypothesized would more accurately index differences in Blue color perception than perceptual alternation rate. Interestingly, male OUD subjects exhibited decreased dwell time for perceiving the Blue stimulus while female OUD subjects evinced increased dwell time for perceiving the Blue stimulus compared to controls. Analysis of the time spent perceiving Blue and Red (i.e., Blue/Red ratio) was decreased in males with OUD, but not females (sex × OUD interaction: *F*_(1,64)_ = 10.19, *p* = 0.0022, partial *η*^2^
*=* 0.1373 [0.0194, 0.2920]; male control vs. male OUD: *t* = −2.65, *p* = 0.020; female control v female OUD: *t* = 1.84, *p* = 0.140; male control vs. female control: *t* = −2.25, *p* = 0.056; male OUD vs. female OUD: *t* = 2.27, *p* = 0.053). There were no further significant effects of sex or opioid use disorder on the ratio (main effect of sex: *F*_(1,64)_ = 0.01, *p* = 0.9353; main effect of OUD: *F*_(1,64)_ = 0.44, *p* = 0.5079).

### 3.6. Effects of Sex and OUD on Neuropsychological Tests

We evaluated various neuropsychological indices of cognition in non-OUD vs. OUD subjects including *Finger Tapping, Symbol Digit Modalities Test* (Symbol Digit Score and Symbol Digit Error), and Trails A/B for sex and OUD effects (Figure 5). For *Finger Tapping* tests, there was a marked difference between non-OUD males vs. females for left finger tapping (*F*_(1,30)_ = 24.1, *p* = 3.3 × 10^−5^; Male non-OUD = 51.2 ± 0.92 taps; Female non-OUD = 42.2 ± 1.56 taps; Figure 5A) and right finger tapping (*F*_(1,29)_ = 17.5, *p* = 0.0003; Male non-OUD = 55.7 ± 0.99 taps; Female non-OUD = 46.9 ± 1.75 taps; Figure 5A). There was a marked difference in both left (*F*_(1,29)_ = 11.7, *p* = 0.002; Male OUD = 49 ± 1.25 taps; Female OUD = 43.6 ± 0.98 taps; Figure 5A) and right (*F*_(1,29)_ = 14.9, *p* = 0.0005; Male OUD = 54.1 ± 1.13 taps; Female OUD = 48.2 ± 1.03 taps; Figure 5A) finger tapping between male and female OUD subjects, but not within sex for non-OUD vs. OUD subjects (*p* > 0.05; Figure 5A).

For the *Trails A/B* tests, there was no significant difference between non-OUD males vs. non-OUD females for *Trails A* (*p >* 0.05; Male non-OUD = 18.8 ± 2.4 s; Female non-OUD = 19.7 ± 1.62 s; Figure 5B) nor for *Trails B* (*p >* 0.05; Male non-OUD= 63.5 ± 0.10.1 s; Females = 44.6 ± 3.77 s; Figure 5B), or for *Trails A* (*p* > 0.05; Male OUD = 27.3 ± 2.6 s; Female OUD = 25.1 ± 1.9 s; Figure 5B) nor *Trails B* (*p* > 0.05; Male OUD = 70.6 ± 9.1 s; Female OUD = 73.5 ± 9.44 s; Figure 5B) between male and female OUD subjects. However, there was a within sex non-OUD controls vs. OUD significant difference for *Trails A* males (*F*_(1,15)_ = 4.8, *p* = 0.05; Figure 5B) and *Trails A* females (*F*_(1,17)_ = 4.63, *p* = 0.05; Figure 5B), and Trails B males (*p* > 0.05), but there was a non-OUD vs. OUD Trails B difference for females (*F*_(1,17)_ = 14.9, *p* = 0.03; Figure 5B).

For the *Symbol Digits Modalities Test (SMDT)*, there was no significant difference between non-OUD males vs. females for *SMDT* scores (*p >* 0.05; Male non-OUD = 62.8 ± 3.2 s; Female non-OUD = 61.4 ± 2.5 s; Figure 5C) or between male vs. female OUD subjects (*p* > 0.05; Male OUD = 48.8 ± 1.97 s; Female OUD = 52.4 ± 1.8 s; Figure 5C). However, there was a within sex non-OUD vs. OUD significant difference for *SMDT* score for males (*F*_(1,33)_ = 15.03, *p* = 0.0004; Male OUD = 48.8 ± 1.97; Figure 5C) and females (*F*_(1,34)_ = 8.87, *p* = 0.005; Figure 5C). There was no significant difference between non-OUD males vs. females for *SMDT Error* (*p >* 0.05; Male non-OUD = 1.0 ± 0.29 errors; Female non-OUD = 0.88 ± 0.22 errors; Figure 5D) or between male vs. female OUD subjects (*p* > 0.05; Male OUD = 1.26 ± 0.37 errors; Female OUD = 0.95 ± 0.31 errors; Figure 5D), or within sex non-OUD vs. OUD subjects (*p* > 0.05; Figure 5D).

Given these sex and OUD differences in color processing, we ran a SAS/STAT^®^ Proc Mixed statistical analysis, which was intended to fit multilevel and hierarchical linear models. The “mixed” statistical data from the current study were nested within naturally occurring hierarchies such as males and females and OUD. Table 4 summarizes the outcome from that analysis. Proc Mix analysis found significance between non-OUD vs. OUD subjects for *Trails A/B*, and marked significance for the *Symbol Digits Task* irrespective of sex. Proc Mix analysis found marked significance between male vs. female non-OUD subjects for *Finger Tapping*. Proc Mix analysis found no significant differences between male vs. female OUD subjects for any test. Proc Mix analysis revealed a mild difference in *Trails A* and marked difference in the *SMDT task* between male non-OUD vs. OUD subjects. Proc Mix analysis also revealed a mild difference in *Trails A/B* and in the *SMDT* task between female non-OUD vs. OUD subjects.

## 4. Discussion

The primary hypothesis of this study was that color visual processing would differ significantly between men and women and between non-OUD controls vs. OUD subjects, similar to what we have recently demonstrated for sex and OUD differences in a visual object recognition task [33]. It was further hypothesized that blue color processing, which is modulated by DA release from amacrine cells in the indirect pathway to the direct blue cone pathway in the retina [39], is a potential biomarker of the addictive state in men and women. The color recognition task was valid in that we were able to distinguish distinct amplitudes and latencies across all VEP and ERP components associated with the Relevant stimulus as in the prior visual object recognition study. Color processing appears to be sex-sensitive and perhaps stimulus-selective (i.e., Blue color Relevant stimuli evoked a robust P300 while Irrelevant Red or Green did not in this Go/No-Go task). As hypothesized, there were significant sex differences found in ERPs from the color recognition task in that male non-OUD controls had greater P300 and LP amplitudes and shorter latencies than female controls—a similar sex difference was also seen in the OUD cohort as hypothesized. We found robust N200 and P300 sex differences in non-OUD subjects exclusively when Blue was the Relevant stimulus. Moreover, as hypothesized, there were sex and OUD ERP differences in amplitudes and latencies for N200, P300, and LP—men and women differed significantly overall and men with OUD had significantly lower P300s than men with no OUD, as observed in other SUDs. Relatively little is known regarding the functional significance of the LP. The LP is most pronounced around 400–800 ms following the stimulus but can last up to a second past the stimulus and likely encompasses motor and response-reflective components and not just decision processing. Brown and colleagues [43] reported some evidence of decreased N100 and P100 amplitudes that accompanied large LPs elicited by unpleasant visual stimuli (i.e., spider pictures), which may indicate the LP reflects a global inhibition of activity in the visual cortex.

We hypothesized that physiological and behavioral indices of color processing would be affected by sex and by OUD. Regarding physiological studies, we evaluated blue color processing by using the *Binocular Rivalry Test*. While there were no sex differences in this test, we found that the time spent dwelling on the Blue stimulus was significantly reduced in subjects with OUD, suggesting that blue color processing in the retina was deficient in OUD, at least in males, perhaps via deficits in amacrine cell DA modulation of the blue cone direct retinal pathway. Of perhaps most importance and significance, we found marked differences in sex and OUD in the results of the neuropsychological tests: *Finger Tapping*, *Symbol Digit Modalities Test*, and the *Trails A/B Test.* These results provide more evidence that chronic opioid use can lead to deficits in attention, memory, and executive functions such as planning and decision-making, and such deficits are different for men and women. It also appears that these particular tests are sensitive and able to pinpoint these specific areas of weakness in cognition as being different in men vs. women in OUD. The tests used for this current color study revealed significant sex-related differences in domains like memory, executive functions (e.g., decision-making for Go/No-Go tasks), and flexibility. These are cognitive functions which are crucial for recovery and further prevention of relapse and can help clinicians distinguish between long-term deficits and temporary effects of acute drug use. Traditionally, there has been little discussion or research regarding the implications of sex differences in the neural substrates that may worsen the effects of certain drugs for men as opposed to women [44,45]. When paired with electrocortical activity, ongoing neuropsychological testing can also be used to track cognitive recovery over time and evaluate the effectiveness of treatment interventions in improving cognitive functions essential for rehabilitation and help inform the patient and clinician to maintain the recovery from any SUD. With such diversity in high risk or predisposition for opioid dependence and addiction, and the alarming evidence that the overdose death rate in young males is currently over twice that of young females, there is clearly a great need for developing more thorough, *individualized* treatment programs for any SUD patients that recognize such sex-related differences.

Potential limitations of this study may come from the smaller sample size for the OUD group. However, very few of the EEG studies on substance abuse have reported large sample sizes; this may be due to the stigma of substance abuse and addiction. Further, similar significant sex differences for the control group in the color discrimination task were found, allowing for increased reliability of the results in the smaller clinical group. Another confound comes from the paucity of EEG research specific to opiate dependence and OUD [30] and color processing [46,47,48,49,50] for designing our experiment to replicate or follow. However, this study does have similar results to those of alcohol and cocaine abuse studies with decreased P300 amplitudes and increased latencies found in OUD [44,45]. Further, the fact that addicts were in withdrawal and active detoxification may limit our results in regard to the effects of “kindling” or hypersensitivity of the limbic system as seen in chronic alcohol withdrawal [51,52]. Nevertheless, our findings did match the literature for changes in ERP data with chronic drug use and alcohol and detoxification [23,53,54,55]. Similar decreased P300 amplitudes and increased latencies related to disinhibition maintained across time (as seen in alcoholism and other chronic drug use) were found. The neural underpinnings of the P300 are poorly understood despite its well-known reflection of high-order cognitive processes including attention allocation, working memory, and stimulus evaluation. As described in this study, there appears to be slowing along the visual pathway, as reflected in latency increases in non-OUD females and subjects with OUD, even in this simple Go/No-Go color recognition task. We have also shown this in a prior study with a simple Go/No-Go visual object recognition task [33]. We are not in a position to speculate where along the visual pathway from the retina to striate cortex or the dorsal/ventral streams that slowing in transmission might be occurring. Indeed, the slowing could be in the retina, as hypothesized. However, P300 is ultimately cortical and increased latency reflects physiological slowing in synaptic transmission along the visual pathway to the cortex, perhaps particularly parietal, frontal, temporal, and even hippocampal cortices. One might speculate that DA improves transmission in the visual pathway given its role in attention, error processing, and learning, in particular in the pre-frontal cortices and hippocampus with known DAergic projections. The dogma is that DA transmission is lower in subjects with OUD, which one might speculate is due to a deficit in the retina or in its modulation of the pre-frontal cortex. In view of the findings of the current color study and the recent sex differences studies for object recognition tasks [34,35,36], there needs to be an increase in the consideration of possible hormonal effects on DA neurotransmission in men and women, including those associated with all types of addiction or other disinhibitory disorders [56]. In conclusion, the results of this study indicate the following:N200, P300, and late potential (LP) *Relevant* stimulus-induced ERPs were evoked by the simple color processing Go/No-Go task and were well-differentiated from *Irrelevant* distractor stimuli.P300 amplitudes were significantly greater and N200 and LP latencies significantly shorter in male vs. female non-OUD controls in this task.N200, P300, and LP ERP amplitudes and/or latencies were significantly affected to varying degrees by sex and OUD, but most significance was found with the LP, latencies, and blue color, some at *p* < 0.0001 levels of significance.In the *Binocular Rivalry Test*, there were shorter dwell times for perceiving a blue stimulus in male OUD subjects.There were significant sex and OUD differences in neuropsychological tests including *Finger Tapping*, *Trails A/B*, and *Symbol Digit Modalities Test*.Although these findings are remarkable and significant, these measures may not be reliable as effective indices for a biomarker of brain DA in SUD, as an effective biomarker must demonstrate both exceptional analytic reliability, reproducibility, and accuracy.Our study provides compelling evidence that the simple color discrimination task used in the current study is sensitive to sex and OUD differences and may provide the basis for the development of an effective biomarker for better understanding the biological underpinnings of substance abuse addiction and as an objective index of mesolimbic DA transmission and monitoring of treatment efficacy.

## Figures and Tables

**Figure 1 biomedicines-13-03002-f001:**
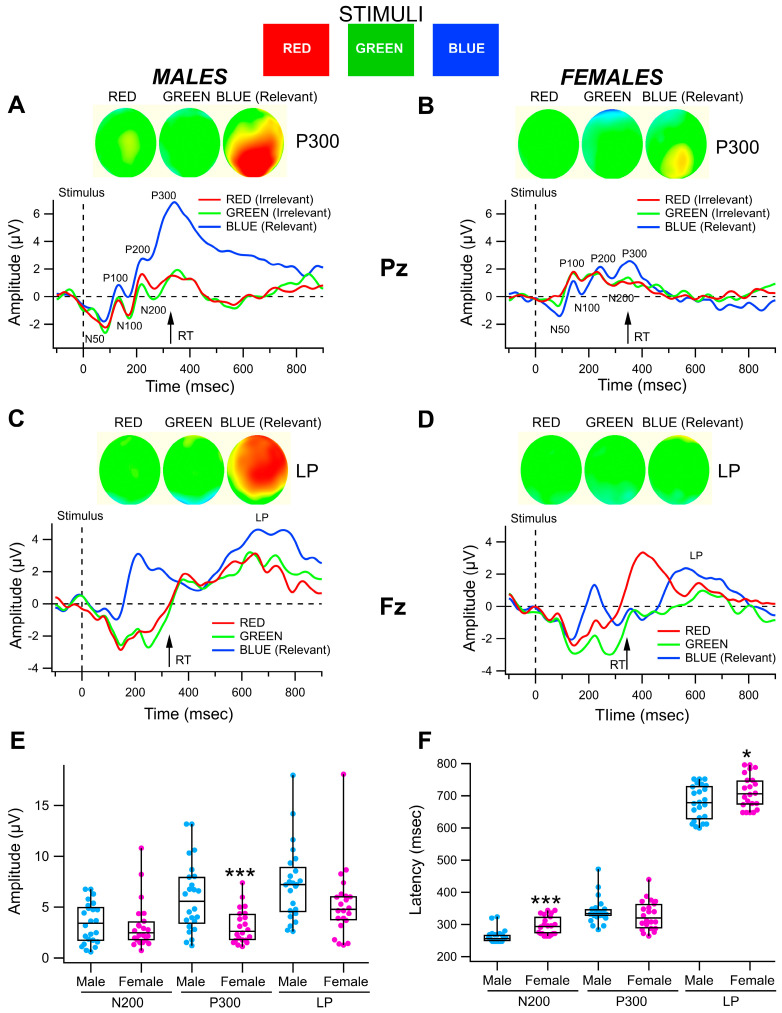
*Sex differences in event-related potentials in a simple Go/No-Go color recognition task in non-OUD control participants*. Red, Green, and Blue stimuli were randomly presented at 2–4 s intervals. Participants responded to the Relevant stimulus, in this case Blue. (**A**,**B**) These graphs show grand-averaged VEPs recorded in male and female non-OUD participants at electrode Pz in response to Red, Green, and Blue stimuli. The vertical scales are normalized to facilitate comparisons between gender. The dashed lines indicate the time of presentation of the stimulus. The components of the VEP included the N50, P100, N100, P200, N200, and P300. Reaction time (RT) is shown with an arrow for the averaged Go response to the Relevant stimulus. Note that this simple color recognition task produced P300s differentiated to Relevant vs. Irrelevant stimuli. The 128 sensor topomaps (insets) represent grand averaged potentials in male and female non-OUD participants at 349 ms (P300) after the presentation of the color stimuli. The color map scales are also normalized to facilitate comparisons between gender for non-OUD participants in this figure. The top of each oval is the front of the head and the bottom of each oval represents the back of the head, as if looking down on the head from above. Violet represents extreme negative potentials and red represents extreme positive potentials. (**C**,**D**) These graphs show grand-averaged VEPs recorded in male and female non-OUD participants at electrode Fz in response to Red, Green, and Blue stimuli, with Blue as the Relevant stimulus. Note that this task induced a late positive (LP) potential associated with the Relevant stimulus that was differentiated from Irrelevant stimuli. The 128 sensor topomaps (insets) represent grand averaged potentials in male and female non-OUD participants at 699 ms (LP) after the presentation of the color stimuli. Average RT is shown for the Relevant stimulus. Note that females have a less prominent LP than males. (**E**,**F**) These box plot graphs summarize N200, P300, and LP ERP amplitude and latency measurements for all male (*n* = 24) and female (*n* = 22) non-OUD participants by gender and ERP component. Male non-OUD subjects were characterized by significantly larger P300 and LP amplitudes and shorter N200 and LP latencies than female non-OUD subjects. Asterisks *, *** indicate significance levels *p* < 0.05 and *p* < 0.001, respectively.

**Figure 2 biomedicines-13-03002-f002:**
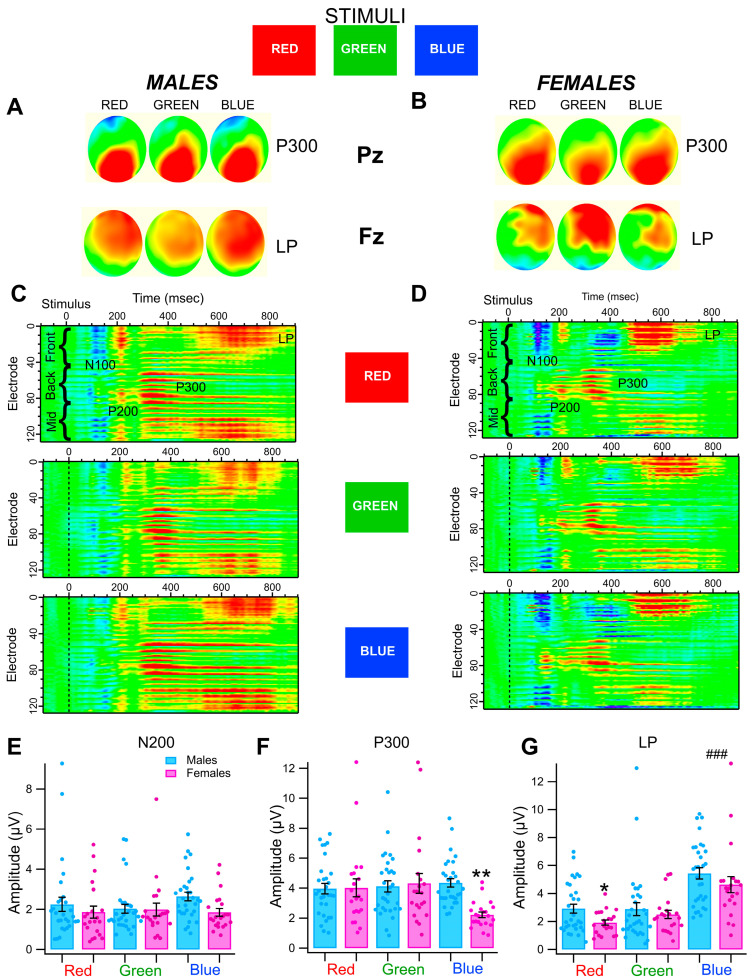
*Effects of wavelength on event-related potentials in male and female non-OUD control participants in the color recognition task*. Participants responded to the *Relevant* stimulus for each of the wavelengths Red, Green, and Blue in separate Go/No-Go experiments. The vertical scales and color maps are NOT normalized in this figure in order to compare Relevant responses for Red, Green, and Blue within sex. (**A**,**B**) Topomaps show grand-averaged P300s (Pz) and LPs (Fz) on the head sensor net in males and females. There was no obvious effect of wavelength on P300s or LPs in non-OUD participants. (**C**,**D**) Synoptic plots showing a heat map of negative (blue) and positive (red) voltage for all 128 electrodes on the head corresponding to the Relevant stimulus for Red, Green, and Blue. It is evident that there were slight differences in the amplitudes of P300 and LP by wavelength. Color plots were normalized within sex for wavelength. (**E**–**G**) Summary of sex differences for the effects of wavelength on ERP amplitudes. There were significant differences between male vs. female P300 amplitudes for Blue as the Relevant stimulus, but not for Red or Green. There was also a significant effect of wavelength on LPs for Red as the Relevant stimulus. Asterisks * and ** indicate significance levels *p* < 0.05 and 0.001 respectively between males and females, and hashtags ### indicate significance level *p* < 0.001 between colors.

**Figure 3 biomedicines-13-03002-f003:**
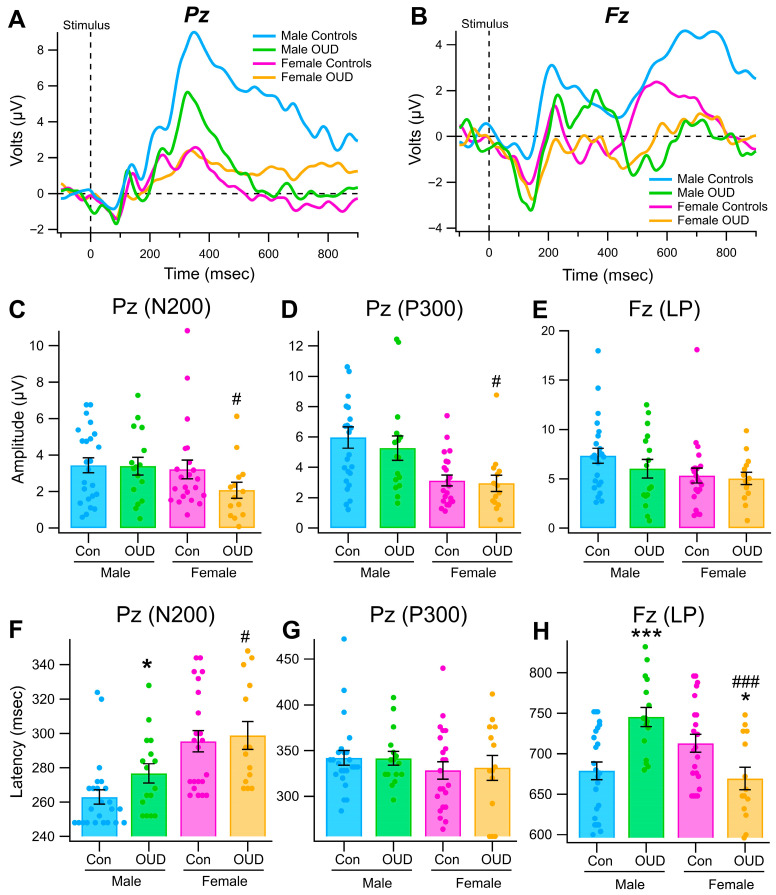
*Event-related potentials in a simple visual recognition task in OUD subjects by gender.* (**A**,**B**) These graphs show grand-averaged waveforms obtained at Pz in both males and females. The vertical scales are NOT normalized in this figure in order to compare Relevant responses for control vs. OUD within gender. Compared to controls, OUD subjects were characterized by smaller N200 and P300s at Pz on the grand-averaged waveform (Relevant stimulus only shown) in both males and females. (**C**,**D**) These graphs show grand-averaged waveforms obtained at Fz in both males and females. The vertical scales are NOT normalized in this figure in order to compare Relevant responses for control vs. OUD within gender. Similarly to Pz, OUD subjects were characterized by reduced LP amplitudes at Fz. (**E**,**F**) These graphs show all the ERP amplitude replicates in male and female controls and OUD for the P300 at Pz and LP at Fz. These are individual amplitude measurements within-subject. Note that male OUD subjects showed smaller P300 amplitudes. (**G**,**H**) These graphs show all the ERP latency replicates in male and female controls and OUD for the P300 at Pz and LP at Fz. These are individual amplitude measurements within-subject. Note that while P300 latencies did not seem to be affected in OUD subjects, LP latencies were significantly longer in males but shorter in females. Asterisks *, *** indicate significance levels *p* < 0.05 and *p* < 0.001 within sex and hashtags #, ### indicate significance levels between sex, respectively.

**Figure 4 biomedicines-13-03002-f004:**
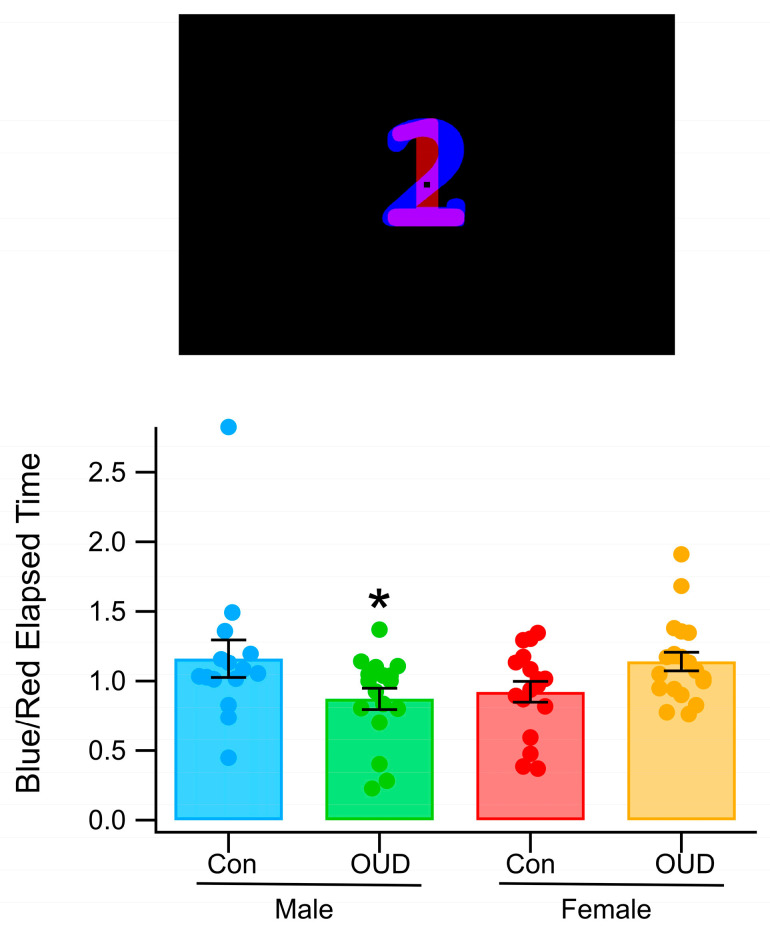
*Sex and OUD differences in blue color processing in binocular rivalry.* The image above the graph shows the superimposed number “1” in red and “2” in blue. When viewed through Red/Blue anaglyph glasses perception alternates between the 2 stimuli at intervals termed binocular rivalry. Subjects were instructed to press the corresponding key on the computer keyboard when they perceived “1” vs. “2”. The graph below shows the ratio of the Blue vs. Red perception elapsed time for all replicates. Male OUD subjects were characterized by less time in the Blue perception compared to Red. Asterisk * indicates significance level *p* < 0.05.

**Figure 5 biomedicines-13-03002-f005:**
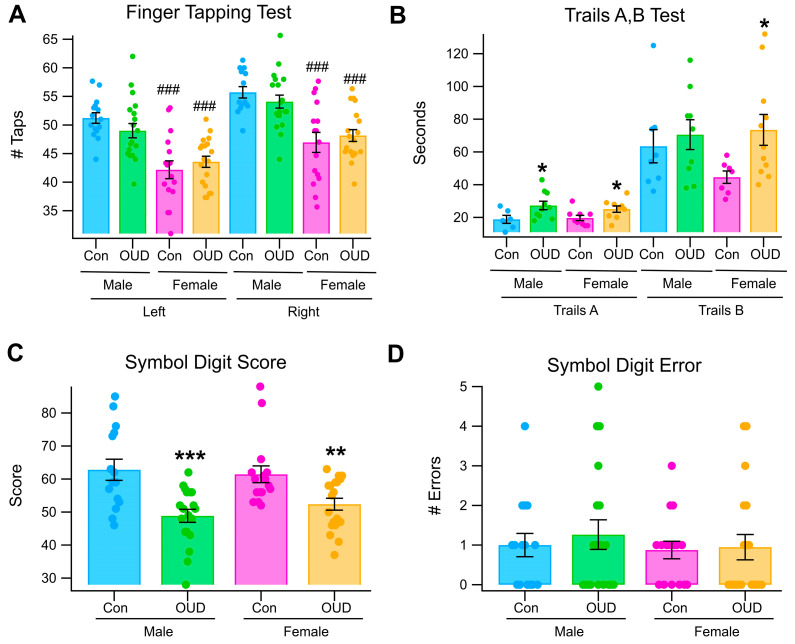
*Sex and OUD differences in neuropsychological tests.* (**A**) This graph shows differences between males vs. females in non-OUD and OUD subjects for all replicates in the Finger Tapping Test. There was marked significance between males vs. females, but no OUD differences. (**B**) This graph shows differences between males vs. females in non-OUD and OUD subjects for all replicates in the Trails A/B Test. There were significant differences between non-OUD and OUD in this test for both males and females. (**C**) This graph shows differences between males vs. females in non-OUD and OUD subjects for all replicates for Symbol Digit Score. There were significant differences between non-OUD and OUD subjects in this test for both males and females. (**D**). This graph shows differences between males vs. females in non-OUD and OUD subjects for all replicates for Symbol Digit Error. There were no sex or OUD differences. Asterisks *, **, *** indicate significance levels *p* < 0.05, *p* < 0.01, and *p* < 0.001 by OUD, hashtags # indicates number of taps, and hashtags ### represent significant differences in errors made in comparisons between sex.

**Table 1 biomedicines-13-03002-t001:** Amplitude of event-related potentials for sex and relevant color in non-OUD controls.

		Color Wavelength		
ERP	Sex	Red M (SD)	Green M (SD)	Blue M (SD)
N200	Male	2.25 (1.94) *n* = 30	2.03 (1.24) *n* = 31	2.64 (1.20) *n* = 31
	Female	1.87 (1.38) *n* = 21	1.99 (1.47) *n* = 21	1.85 (0.90) *n* = 22
P300	Male	3.96 (1.89) *n* = 30	4.12 (2.03) *n* = 31	4.34 (1.55) *n* = 31
	Female	4.02 (2.72) *n* = 21	4.31 (3.02) *n* = 21	2.22 (0.87) *n* =22 **
LP	Male	2.92 (1.80) *n* = 31	2.88 (2.56) *n* = 31	5.43 (2.26) *n* = 31
	Female	1.92 (0.84) *n* = 21 ***	2.50 (1.33) *n* = 21 ***	4.63 (2.66) *n* = 22 ***

***Note:*** ERP = event-related potentials. ** *p* < 0.01, *** *p* < 0.001.

**Table 2 biomedicines-13-03002-t002:** Degree of significance of sex and target color on the amplitude of event-related potentials in non-OUD controls in the color recognition task.

ERP	Factors	*t*	*p*	*b*	95% CI
N200	Sex: female	−0.75	0.455	−0.37	[−1.36, 0.62]
	Color: green	−0.63	0.534	−0.22	[−0.94, 0.49]
	Color: blue	0.92	0.361	0.39	[−0.46, 1.24]
	Interaction: female × green	0.60	0.554	0.33	[−0.79, 1.46]
	Interaction: female × blue	−0.63	0.533	−0.42	[−1.77, 2.89]
P300	Sex: female	0.12	0.904	0.08	[−1.22, 1.38]
	Color: green	0.38	0.706	0.20	[−0.84, 1.23]
	Color: blue	0.85	0.401	0.42	[−0.58, 1.42]
	Interaction: female × green	0.16	0.875	0.13	[−1.49, 1.75]
	**Interaction: female × blue**	**−2.79**	**0.007**	**−2.20**	**[−3.78, −0.62]**
LP	**Sex: female**	**−2.22**	**0.031**	**−0.95**	**[−1.80, −0.09]**
	Color: green	−0.07	0.945	−0.03	[−0.96, 0.90]
	**Color: blue**	**5.85**	**<0.001**	**2.51**	**[1.65, 3.38]**
	Interaction: female × green	0.76	0.451	0.56	[−0.91, 2.02]
	Interaction: female × blue	0.21	0.833	0.15	[−1.25, 1.54]

***Note:*** ERP = event-related potentials; b = slope, CI = confidence interval. Bolded items are significant.

**Table 3 biomedicines-13-03002-t003:** Effects of sex and OUD on event-related potential amplitude and latency in the color recognition task.

		AMP		LAT	
ERP	Contrasts	*F*	*p*	*F*	*p*
N200	Men vs. Women Controls	51.8	<0.0001 ***	96.33	<0.0001 ***
	Controls vs. OUD	0.94	0.33	5.1	0.02 *
	Men Controls vs. Men OUD	4.62	0.03 *	6.7	0.01 *
	Women Controls vs. Women OUD	0.44	0.5	0.47	0.49
P300	Men vs. Women Controls	123.2	<0.0001 ***	2.3	0.13
	Controls vs. OUD	2.61	0.11	1.15	0.29
	Men Controls vs. Men OUD	8.95	0.003 **	5.5	0.02 *
	Women Controls vs. Women OUD	0.3	0.57	0.5	0.48
LP	Men vs. Women Controls	58.1	<0.0001 ***	43.2	<0.0001 ***
	Controls vs. OUD	2.3	0.13	39.4	<0.0001 ***
	Men Controls vs. Men OUD	1.92	0.17	288.25	<0.0001 ***
	Women Controls vs. Women OUD	0.59	0.44	50.7	<0.0001 ***

***Note:*** ERP = event-related potentials; AMP = amplitude; LAT = latency; OUD = opioid use disorder. Degrees of freedom for all values = 450; * *p* < 0.05, ** *p* < 0.001, *** *p* < 0.0001.

**Table 4 biomedicines-13-03002-t004:** Neuropsychological testing comparisons for sex and OUD.

Tests	Group Comparisons				
	Controls M (SD) (*n* = 31) ± vs. Addiction M (SD) (*n* = 38)	FC M (SD) (*n* = 16) vs. MC M (SD) (*n* = 15)	FA M (SD) (*n* = 19) vs. MA M (SD) (*n* = 19)	MC M (SD) (*n* = 15) vs. MA M (SD) (*n* =19)	FC M (SD) (*n* = 16) vs. FA M (SD) (*n* = 19)
Finger Tapping	51.3 (7.2) 51.1 (7.2)	46.9 (7.0) 55.9 (3.7) ***	48.2 (4.5) 54.1 (4.9)	55.9 (3.7) 54.1 (4.9)	46.9 (7.0) 48.2 (4.5)
Trails A	19.3 (5.2) (*n* = 15)	19.7 (4.9) (*n* = 6)	25.1 (5.8) (*n* = 9)	18.8 (6.0) (*n* = 9)	19.7 (4.9) (*n* = 6)
	26.3 (7.1) * (*n* = 19)	18.8 (6.0) (*n* = 9)	27.3 (8.2) (*n* = 10)	27.3 (8.2) * (*n* = 10)	25.1 (5.8) * (*n* = 9)
Trails B	54.7 (23.4) (*n* = 15)	44.6 (10) (*n* = 7)	73.5 (31.3) (*n* = 11)	63.5 (28.6) (*n* = 8)	44.6 (10) (*n* = 7)
	72.2 (28.8) * (*n* = 20)	63.5 (28.6) (*n* = 8)	70.6 (27.3) (*n* = 9)	70.6 (27.3) (*n* = 9)	73.5 (31.3) * (*n* = 11)
Symbol Digit Modalities Test	50.6 (8.3) 62.1 (11.1) ***	61.4 (10.2) 62.8 (12.4)	52.4 (7.8) 48.8 (8.6)	62.8 (12.4) 48.8 (8.6) ***	61.4 (10.2) 52.4 (7.8) *

***Note****:* M = mean; SD = standard deviation; FC = women controls; FA = women addicts; MC = male controls; MA = male addicts. * *p* < 0.05, *** *p* < 0.0001. ± = the *n* values for the Trails tests are different—we alternated the tests allowing for second testing and no test-retest confounds.

## Data Availability

All data generated or analyzed during this study are included in this published article and its Appendix A. It is available on request from SCS.

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
