# Peer review of "Sex-Specific Electrocortical Interactions in a Color Recognition Task in Men and Women with Opioid Use Disorder"

_biomedicines, 2025, doi:10.3390/biomedicines13123002_

Round 1

Reviewer 1 Report

Comments and Suggestions for Authors

This manuscript addresses a highly relevant and timely issue regarding sex-specific neural mechanisms in opioid use disorder (OUD), focusing on electrophysiological indices (EEG/ERP) during a color recognition task. The study design is commendable for integrating behavioral, perceptual, and neurophysiological measures to explore dopamine-related visual processing differences. The inclusion of both male and female participants and the use of a color-based Go/No-Go paradigm are original and potentially impactful.

However, while the topic is compelling and the methodology is rich in detail, the manuscript would benefit from substantial refinement in structure, clarity, and statistical interpretation. The current manuscript presents an extensive methodological description but lacks concise articulation of hypotheses, a clear connection between ERP findings and the proposed dopaminergic mechanism, and a more rigorous statistical reporting format. The discussion section should better situate findings within existing neurobiological and clinical frameworks of OUD.

Major Comments

  1. The introduction provides comprehensive background information but should more explicitly define the primary hypothesis and secondary objectives. For example, the rationale linking blue color processing to dopaminergic function is interesting but currently reads as speculative rather than hypothesis-driven.

  1. The Methodology section is comprehensive but lacks a coherent and logical flow. The current order mixes participant recruitment details, EEG setup procedures, and behavioral task descriptions in a way that makes it difficult to follow. To improve readability and scientific rigor, the section should be reorganized into a clear, hierarchical structure with consistent subheadings.

  1. While the Results section presents a large volume of electrophysiological and behavioral data, its interpretation lacks clarity and narrative coherence.

  1. The statement “An important strength of the current color study is our inclusion of physiological and neuropsychological testing” may sound overly self-promotional.

  1. The Discussion section is relatively underdeveloped, providing limited theoretical integration and insufficient linkage to the broader neurobiological and clinical context of opioid use disorder (OUD). Conversely, the Conclusion section is disproportionately long and somewhat repetitive, reiterating methodological details and minor findings.

Author Response

We greatly appreciate the thoughtful comments and suggestions of the reviewers.  We feel that the study is improved by their feedback. We are grateful for the opportunity to revise and resubmit for publication in Biomedicines.  We respond below to the reviewer’s comments.

 Rev 1

This manuscript addresses a highly relevant and timely issue regarding sex-specific neural mechanisms in opioid use disorder (OUD), focusing on electrophysiological indices (EEG/ERP) during a color recognition task. The study design is commendable for integrating behavioral, perceptual, and neurophysiological measures to explore dopamine-related visual processing differences. The inclusion of both male and female participants and the use of a color-based Go/No-Go paradigm are original and potentially impactful.

However, while the topic is compelling and the methodology is rich in detail, the manuscript would benefit from substantial refinement in structure, clarity, and statistical interpretation. The current manuscript presents an extensive methodological description but lacks concise articulation of hypotheses, a clear connection between ERP findings and the proposed dopaminergic mechanism, and a more rigorous statistical reporting format. The discussion section should better situate findings within existing neurobiological and clinical frameworks of OUD.

Major Comments

  1. The introduction provides comprehensive background information but should more explicitly define the primary hypothesis and secondary objectives. For example, the rationale linking blue color processing to dopaminergic function is interesting but currently reads as speculative rather than hypothesis-driven.

Response: We expanded and clarified the hypotheses in the Introduction and included additional references for the rationale.

  1. The Methodology section is comprehensive but lacks a coherent and logical flow. The current order mixes participant recruitment details, EEG setup procedures, and behavioral task descriptions in a way that makes it difficult to follow. To improve readability and scientific rigor, the section should be reorganized into a clear, hierarchical structure with consistent subheadings.

Response: Methods were reorganized in a logical structure.

  1. While the Results section presents a large volume of electrophysiological and behavioral data, its interpretation lacks clarity and narrative coherence.

Response: We have re-organized the Results section in an attempt to improve coherence and flow including segues between experiments with subheadings.

  1. The statement “An important strength of the current color study is our inclusion of physiological and neuropsychological testing” may sound overly self-promotional.

Response: We agree.  We have removed this statement.

  1. The Discussion section is relatively underdeveloped, providing limited theoretical integration and insufficient linkage to the broader neurobiological and clinical context of opioid use disorder (OUD). Conversely, the Conclusion section is disproportionately long and somewhat repetitive, reiterating methodological details and minor findings.

Response: We have shortened the Conclusion paragraph with important bulleted points.

Reviewer 2 Report

Comments and Suggestions for Authors

The article under review is dedicated to the investigation of opioid use disorder (OUD) on color processing in “males” and “females”. Authors recently showed sex-specific difference in object recognition upon OUD and now they claim that the sex-specific difference in color recognition may be used as diagnostic marker and may be linked with increased mortality in men in comparison to women.

The article is interesting and important but written in very complex, difficult language, which makes it difficult to understand the data. Some details should be clarified.

Introduction is fine

Materials should be supplemented

Results should be made more concise and clearer

Discussion and conclusions can be united

Conclusions should be rewritten

Declarations are excellent

References are fine (maybe reformatting is needed)

Figures are fine

My points.

  1. The results are hard to follow (digits in brackets are lines).

E.g. part 3.1: Methods-methods (378-391) – results (391-409) – electrodes (409-413, the repetition of stat analysis) – results (413-418) – results-statistics. It is hard to follow and quite hard to convert into physiology. I advise you to rearrange the material in this way:

- Put 378-391, 409-413 in the methods. Also, it would be great to provide us such table (for all Eps analyzed)

Table X. The EEG data collected within the study  

VEP

Electrode

Scalp areas

Related process

Visual evoked potentials (VEPs)

Early

N50

P100

Late

N200

T5, P3, Pz, P4, T6, O1 and O2

frontal and central

mismatch detection, error monitoring, executive control

P300

LP

FP1, FP2, F7, F3, Fz, F4, 411 and F8

  • Start the results with line 391, delete 413-418. Try to remove any methodological details to methods section.
  • 418 – “ANOVA at the PZ electrode” sounds unusual, please rephrase in all parts of the manuscript (maybe “data analysis showed that that at PZ electrode males had significantly greater 418 P300 amplitude than females”).
  • Figure legends – provide the details of statistical tests used (see later).
  • Generally, the ANOVA description in the results “eclipses” the physiological side of effects. Consider repeating main findings after ANOVA description. Maybe, put some concluding sentence in the end of the part (move lines 454-455 from the figure legend to line 430). This is applicable to all parts, especially when the ANOVA’s “interaction” appears.

For me, part 3.3. is more “understandable” than other ones.

  1. Please describe the ANOVA in methods in more detail. First show normality tests used, what kind of ANOVA was used (Welch-ANOVA if I’m not mistaken), what post hoc test was used (Games-Howell?).
  2. Put the conclusion part in the discussion and make conclusions more concise and shorter. Maybe as bullets:

Conclusions:

  1. Non-OUD males demonstrate higher P300 amplitude but lower N200 and LP latency in EEG during Go/No-Go color recognition task.
  2. Both males and females demonstrate the higher LP amplitude for blue color recognition. Females demonstrate shorter P300 and LP amplitude in recognition of blue and red colors, respectively.
  3. And so on…
  4. Abstract is too long, please shorten it maybe using “conclusions bullets”

Minor:

  1. Increase the font size on the figures
  2. Check the references style (are doi required?)

Generally, the article is very good, but it should be revised so that the reader can “grab the message” quickly.

I recommend major revision.

Comments on the Quality of English Language

The language is very good but more "physiological aspects" of the results should be added

Author Response

We greatly appreciate the thoughtful comments and suggestions of the reviewers.  We feel that the study is improved by their feedback. We are grateful for the opportunity to revise and resubmit for publication in Biomedicines.  We respond below to the reviewer’s comments.

Rev 2

The article under review is dedicated to the investigation of opioid use disorder (OUD) on color processing in “males” and “females”. Authors recently showed sex-specific difference in object recognition upon OUD and now they claim that the sex-specific difference in color recognition may be used as diagnostic marker and may be linked with increased mortality in men in comparison to women.

The article is interesting and important but written in very complex, difficult language, which makes it difficult to understand the data. Some details should be clarified.

Introduction is fine

Materials should be supplemented

Response: There are very few materials in the Methods section.  We are not aware of supplemental documents that are used in the journal for this. 

Results should be made more concise and clearer

Response: We have made a concerted effort to clarify the Results (see below).

Discussion and conclusions can be united

Done

Conclusions should be rewritten

Conclusions have been abbreviated as the last paragraph of the Discussion and summarized with important bulleted points.

Declarations are excellent

References are fine (maybe reformatting is needed)

Figures are fine

My points.

  1. The results are hard to follow (digits in brackets are lines).

Response: Sorry, we are not sure what the phrase “digits and brackets are lines” means. Regardless, we have reorganized the Results section as per Rev 1

E.g. part 3.1: Methods-methods (378-391) – results (391-409) – electrodes (409-413, the repetition of stat analysis) – results (413-418) – results-statistics. It is hard to follow and quite hard to convert into physiology. I advise you to rearrange the material in this way:

- Put 378-391, 409-413 in the methods. Also, it would be great to provide us such table (for all Eps analyzed)

Response: Thanks for making this recommendation and the suggestions below. Although we started to make a Table as per these suggestions we struggled with where to put it, as it represents a review of the literature and somewhat distracted from our findings. This paper is not a review of the lit. However, we realized that it would be better to explain in the text more about the different ERPs and their significance to cognitive function, which we have done in the revision. Regardless, we appreciate this suggestion by the reviewer.

Table X. The EEG data collected within the study  

VEP

Electrode

Scalp areas

Related process

Visual evoked potentials (VEPs)

Early

N50

P100

Late

N200

T5, P3, Pz, P4, T6, O1 and O2

frontal and central

mismatch detection, error monitoring, executive control

P300

LP

FP1, FP2, F7, F3, Fz, F4, 411 and F8

  • Start the results with line 391, delete 413-418. Try to remove any methodological details to methods section.

Response: Done with some modification.

  • 418 – “ANOVA at the PZ electrode” sounds unusual, please rephrase in all parts of the manuscript (maybe “data analysis showed that that at PZ electrode males had significantly greater 418 P300 amplitude than females”).

Response: We have modified the stats section and taken out reference to “ANOVA at the __ revealed” jargon. Most of the stats in this study were not ANOVA, but a Proc Mix model.

  • Figure legends – provide the details of statistical tests used (see later).

Response, we respectfully disagree that the stats need to be in the captions. They are in the Methods and Results sections.  However, we did add significance level indications in the captions for the asterisks, etc… in the figure graphs.

  • Generally, the ANOVA description in the results “eclipses” the physiological side of effects. Consider repeating main findings after ANOVA description. Maybe, put some concluding sentence in the end of the part (move lines 454-455 from the figure legend to line 430). This is applicable to all parts, especially when the ANOVA’s “interaction” appears.

Response: Sorry, we are not sure what this means. However, we have clarified the use of ANOVA throughout the Results.

For me, part 3.3. is more “understandable” than other ones.

  1. Please describe the ANOVA in methods in more detail. First show normality tests used, what kind of ANOVA was used (Welch-ANOVA if I’m not mistaken), what post hoc test was used (Games-Howell?).

Response: See above

  1. Put the conclusion part in the discussion and make conclusions more concise and shorter. Maybe as bullets:

Response: Thanks for this suggestion.  We have bulleted the important points at the end of the Discussion.  We are not sure why the reviewer does not like the use of the subheading Conclusion, but we have taken it out.

Conclusions:

  1. Non-OUD males demonstrate higher P300 amplitude but lower N200 and LP latency in EEG during Go/No-Go color recognition task.
  2. Both males and females demonstrate the higher LP amplitude for blue color recognition. Females demonstrate shorter P300 and LP amplitude in recognition of blue and red colors, respectively.
  3. And so on…
  4. Abstract is too long, please shorten it maybe using “conclusions bullets”

Minor:

  1. Increase the font size on the figures

Response: We have increased the font sizes for the axes on all graphs in all the figures.  We have also increased the resolution of the figure images to 1200 DPI.

  1. Check the references style (are doi required?)

Response: We are following the same reference style used in our previous publication in Biomedicines.

Generally, the article is very good, but it should be revised so that the reader can “grab the message” quickly.

I recommend major revision.

Round 2

Reviewer 2 Report

Comments and Suggestions for Authors

The authors expanded on their methods and tried to describe the physiological significance of their data.

I still believe the article is overloaded with statistical data interpretations and descriptions of observed parameters without physiological context, but despite this, I can recommend it for publication.

Perhaps a more detailed discussion of the physiological significance of your data would be helpful.

Note how the results are discussed in 10.1038/s41598-023-43132-8 (don't cite this paper, it's just an example), the discussion, 2nd paragraph:

-authors describe their effects first:

“Although other studies showed individuals with MCI have prolonged P300 latencies and smaller P300 amplitudes during visual tasks, our results showed only prolonged P300 latency during target processing in the MCI group compared to their matched controls. It has been previously reported that even healthy aging is associated with decreased P300 amplitude and prolonged P300 latency during a visual task60–63. However, previous work from our lab indicated that P300 amplitudes are affected by age but not the P300 latency during visual oddball paradigm64. In addition to the aging effect on P300 latency, neural correlates of MCI slow down the P300 response even more in the MCI group.”

-authors provide the physiological output of their results:

“What does P300 latency indicate? P300 latency is ONE OF THE MOST COMMON ASPECTS OF THE P300 WAVE THAT IS THOUGHT TO REFLECT POST-STIMULUS INFORMATION PROCESSING32,63,65,66 INCLUDING CLASSIFICATION SPEED32 AND EXECUTIVE FUNCTIONS (ATTENTION, MEMORY)67,68. Prolonged P300 latency in target processing compared to non-target processing in the MCI group and prolonged P300 response in target processing in the MCI group compared to controls indicate slower processing in the MCI group when they need to identify and classify the target stimulus. thus, our results provide evidence that high-order cognitive processes (i.e. executive functioning and memory) that are involved in stimulus processing slow down in individuals with MCI due to the high working memory demand for neural processing”.

Revising the discussion in that way (for every observed difference) may significantly increase the impact of your work, however this is not mandatory. Consider minor revision.

Comments on the Quality of English Language

The language is very good but more "physiological aspects" of the results should be added

Author Response

Response: We greatly appreciate the thoughtful comments and suggestions of Rev 2 on this second revision of our manuscript.  We feel that the study is improved by the reviewer's feedback. We are grateful for the opportunity to revise again and resubmit for publication in Biomedicines.  We respond below to the reviewer’s comments.

The authors expanded on their methods and tried to describe the physiological significance of their data.

I still believe the article is overloaded with statistical data interpretations and descriptions of observed parameters without physiological context, but despite this, I can recommend it for publication.

Perhaps a more detailed discussion of the physiological significance of your data would be helpful.

Response: We have attempted to simplify a very complex dataset. Thus, we have created a table with descriptions of the ERPs and their physiological context, but we don’t feel it should be in the manuscript.  It is review material.  However, we have uploaded the table into the Submission system, but we feel it should be Supplemental material.

Note how the results are discussed in 10.1038/s41598-023-43132-8 (don't cite this paper, it's just an example), the discussion, 2nd paragraph:

-authors describe their effects first:

“Although other studies showed individuals with MCI have prolonged P300 latencies and smaller P300 amplitudes during visual tasks, our results showed only prolonged P300 latency during target processing in the MCI group compared to their matched controls. It has been previously reported that even healthy aging is associated with decreased P300 amplitude and prolonged P300 latency during a visual task60–63. However, previous work from our lab indicated that P300 amplitudes are affected by age but not the P300 latency during visual oddball paradigm64. In addition to the aging effect on P300 latency, neural correlates of MCI slow down the P300 response even more in the MCI group.”

-authors provide the physiological output of their results:

“What does P300 latency indicate? P300 latency is ONE OF THE MOST COMMON ASPECTS OF THE P300 WAVE THAT IS THOUGHT TO REFLECT POST-STIMULUS INFORMATION PROCESSING32,63,65,66 INCLUDING CLASSIFICATION SPEED32 AND EXECUTIVE FUNCTIONS (ATTENTION, MEMORY)67,68. Prolonged P300 latency in target processing compared to non-target processing in the MCI group and prolonged P300 response in target processing in the MCI group compared to controls indicate slower processing in the MCI group when they need to identify and classify the target stimulus. thus, our results provide evidence that high-order cognitive processes (i.e. executive functioning and memory) that are involved in stimulus processing slow down in individuals with MCI due to the high working memory demand for neural processing”.

Response: We appreciate these suggestions.  Yes, we need to speculate more in the Discussion regarding neural correlates for the ERPs and their relevancy to our findings.  Thus, we have modified the end of the Discussion to reflect this recommendation; mainly: “The neural underpinnings of the P300 are poorly understood despite its well-known reflection of high-order cognitive processes including attention allocation, working memory, and stimulus evaluation. As described in this study, there appears to be slowing along the visual pathway, as reflected in latency increases in non-OUD females and subjects with OUD, even in this simple Go/No-Go color recognition task. We have also shown this in a prior study with a simple Go/No-Go visual object recognition task[33]. We are not in a position to speculate where along the visual pathway from the retina to striate cortex or the dorsal/ventral streams that slowing in transmission might be occurring. Indeed, the slowing could be in the retina, as hypothesized. However, P300 is ultimately cortical and increased latency reflects physiological slowing in synaptic transmission along the visual pathway to the cortex, perhaps particularly parietal, frontal, temporal, and even hippocampal cortices. One might speculate that DA improves transmission in the visual pathway given its role in attention, error processing, and learning, in particular pre-frontal cortices and hippocampus with known DAergic projections. The dogma is that DA transmission is lower in subjects with OUD, which one might speculate is due to a deficit in the retina or in its modulation of the pre-frontal cortex.”

Revising the discussion in that way (for every observed difference) may significantly increase the impact of your work, however this is not mandatory. Consider minor revision.